# High Expression of AhR and Environmental Pollution as AhR-Linked Ligands Impact on Oncogenic Signaling Pathways in Western Patients with Gastric Cancer—A Pilot Study

**DOI:** 10.3390/biomedicines12081905

**Published:** 2024-08-20

**Authors:** Martine Perrot-Applanat, Cynthia Pimpie, Sophie Vacher, Marc Pocard, Véronique Baud

**Affiliations:** 1INSERM U1275, Peritoneal Carcimomatosis Paris-Technologies, Hôpital Lariboisiere, Université Paris Cité, 75010 Paris, France; cynthia.crocheray@inserm.fr (C.P.); marc.pocard@inserm.fr (M.P.); 2Department of Genetics, Curie Institute, PSL Research University, 75005 Paris, France; sophie.vacher@curie.fr; 3Department of Digestive and Oncology Surgery, Hôpital Lariboisiere, Université Paris Cité, 75010 Paris, France; 4NF-kappaB, Differentiation and Cancer, Faculty of Pharmacy, Université Paris Cité, 75006 Paris, France

**Keywords:** gastric cancers, aryl hydrocarbon receptor (AhR), diffuse gastric subtype, intestinal gastric subtype, benzo[a]pyrene (BaP), 2,3,7,8-tetrachlorodibenzeno-p-dioxin (TCDD or dioxin), xenobiotic metabolism, tryptophan metabolism

## Abstract

The vast majority of gastric cancer (GC) cases are adenocarcinomas including intestinal and diffuse GC. The incidence of diffuse GC, often associated with poor overall survival, has constantly increased in Western countries. Epidemiological studies have reported increased mortality from GC after occupational exposure to pro-carcinogens that are metabolically activated by cytochrome P450 enzymes through aryl hydrocarbon receptor (AhR). However, little is known about the role of AhR and environmental AhR ligands in diffuse GC as compared to intestinal GC in Western patients. In a cohort of 29, we demonstrated a significant increase in AhR protein and mRNA expression levels in GCs independently of their subtypes and clinical parameters. *AhR* and *RHOA* mRNA expression were correlated in diffuse GC. Further, our study aimed to characterize in GC how AhR and the AhR-related genes cytochrome P450 1A1 (CYP1A1) and P450 1B1 (CYP1B1) affect the mRNA expression of a panel of genes involved in cancer development and progression. In diffuse GC, *CYP1A1* expression correlated with genes involved in IGF signaling, epithelial–mesenchymal transition (*Vimentin*), and migration (*MMP2*). Using the poorly differentiated KATO III epithelial cell line, two well-known AhR pollutant ligands, namely 2-3-7-8 tetrachlorodibenzo-p-dioxin (TCDD) and benzo[a]pyrene (BaP), strongly increased the expression of *CYP1A1* and *Interleukin1β* (*IL1B*), and to a lesser extend *UGT1*, *NQO1,* and *AhR Repressor* (*AhRR*). Moreover, the increased expression of *CYP1B1* was seen in diffuse GC, and IHC staining indicated that CYP1B1 is mainly expressed in stromal cells. TCDD treatment increased *CYP1B1* expression in KATO III cells, although at lower levels as compared to *CYP1A1*. In intestinal GC, *CYP1B1* expression is inversely correlated with several cancer-related genes such as *IDO1*, a gene involved in the early steps of tryptophan metabolism that contributes to the endogenous AhR ligand kynurenine expression. Altogether, our data provide evidence for a major role of AhR in GC, as an environmental xenobiotic receptor, through different mechanisms and pathways in diffuse and intestinal GC. Our results support the continued efforts to clarify the identity of exogenous AhR ligands in diffuse GC in order to define new therapeutic strategies.

## 1. Introduction

GC still remains the fifth leading cause of cancer mortality [1,2,3], with high heterogeneity in different subtypes according to the classification proposed by the World Health Organization [4]. Regardless of the country, the majority of gastric tumors are adenocarcinomas, which can be more histologically classified into diffuse and intestinal subtypes according to the Lauren classification [5]. While intestinal-type GC is well differentiated and related to *Helicobacter pylori* infection, the diffuse GC is poorly differentiated, and can be seen in familial (germline mutation in the E-Cadherin (*CDH1*) gene) or sporadic settings as an infiltrating and scattered type [6,7] with unknown origin. The prevalence of the diffuse type is increasing worldwide, especially in Western countries [8]. The majority of patients with diffuse GC are usually diagnosed at advanced stages with positive axillary nodes (83%) and peritoneal carcinomatosis (18.6%) [6,9,10].

The hypothesis of a link between exposure to persistent organic pollutants (POPs, including dioxin) and cancers, including GC, is supported by several epidemiological studies of accidentally exposed populations in Seveso [11]. Moreover, it is known that polycyclic polyhalogenated hydrocarbons, like dioxins and polychlorobyphenyls (PCBs) and polycyclic aromatic hydrocarbons (PAHs) (including benzo[a]pyrene from tobacco-smoke and biomass burning), are linked to relatively high health risk including cancers. We have previously shown that environmental chemicals, such as POPs are involved in breast cancers [12], and may accumulate in the Omentum adipose tissue of patients with GC [13]. In addition to TCDD, the prototypical and most potent known environmental ligand in animals and humans, other widespread environmental POP contaminants bind AhR, a basic helix-loop-helix transcription factor, with strong affinities [14,15] and may chronically activate AhR in cancer progression.

For decades, AhR has been studied for its role in environmental chemical toxicity and as a mediator of the unintended consequences of human pollution. AhR is a regulator of xenobiotic metabolic enzymes such as CYP1A1 and CYP1B1 [16]. Further, AhR participates in important cellular and pathological processes, including proliferation, migration, angiogenesis, the control of the immune system, and tumor initiation and progression [12,15,17,18,19,20,21,22,23]. Animal experimental data have provided substantial support for an association between AhR and GCs [23,24,25,26,27,28]. Transgenic mice expressing a constitutively active dioxin/AhR mutant (CA-AhR) due to a deletion in the ligand-binding domain of AhR rapidly developed stomach cancers [24,25]. Several tissues of CA-AhR mice have shown the life-long continuous low-level activity of AhR, and this model is useful to mimic the dioxin exposure of humans in the general population [26]. Mice exposed to the chemical carcinogen BaP [27] also developed gastric tumors. There is evidence for elevated AhR expression in human GC [28,29]. However, the expression levels of AhR and AhR-related signaling pathways have not been investigated in Western patients with diffuse GCs.

In this study, we have analyzed the expression of AhR in a series of diffuse and intestinal gastric tumors, as well as the expression of xenobiotic metabolic enzymes such as cytochromes P450 (*CYP1A1* and *CYP1B1*) and a large panel of genes known to be regulated by AhR in several cancers. The impact of exogenous environmental POPs known as AhR ligands on AhR and AhR-related gene expression was also evaluated in vitro in two GC-derived cell lines.

## 2. Materials and Methods

### 2.1. Patients and Tissue Samples

A total of 29 patients underwent partial gastrectomy for histopathologically confirmed gastric adenocarcinoma primary tumor tissue in the Lariboisiere Hospital (Paris, France) from 2005 to 2014. All the patients provided written informed consent prior to their inclusion in the study. Biopsies (provided before 2014) were taken for diagnostic and research purposes, and analysis was permitted by the Ethical Committee of Lariboisiere Hospital (Paris). Eligibility criteria included (1) gastric carcinoma identified by histopathological examination, (2) no other malignancy, (3) no pre-operative chemotherapy or radiotherapy, and (4) the availability of complete clinical, histological, and biological data. Normal (non-malignant) samples refer to the samples harvested from the stomach from sites distant from the tumor. Immediately after surgery, fresh gastric tumors and their matched normal mucosa were stored in liquid nitrogen until mRNA extraction; other tumor samples and their adjacent normal tissues were routinely fixed in 10% buffered formalin and embedded in paraffin for histological analysis. As previously described [10], the population was divided into two groups according to the histological status of GC: diffuse adenocarcinoma (a poorly differentiated, infiltrating, and scattered type) or intestinal adenocarcinoma (a well-differentiated and clustered subtype) according to the Lauren classification (see Table 1). The malignancy of infiltrating carcinomas was scored according to the TNM staging system (stages I to IV) as previously described [10]: first, according to American Joint Committee on Cancer (AJCC) 7 [30], revised from International Gastric Cancer Association [31,32] and AJCC 8 [33]. This TNM staging includes T scores for the primary tumor (T1–T4), N scores (lymph node metastasis), and M scores (metastasis). The clinicopathological characteristics of the 29 patients were already reported in reference [29].

### 2.2. Total RNA Preparation and qRT-PCR

Total RNA extraction, complementary cDNA synthesis, and qRT-PCR conditions were as previously described [12,34]. Primers for *AhR*, *AhRR*, and other genes were selected using the Oligo 6.0 program (National Biosciences, Plymouth, MN) [12,29]. Each sample was normalized on the basis of its *TBP* content as previously described [10,12]. The results, expressed as N-fold difference in target gene expression relative to the *TBP* gene (and termed “*Ntarget*”), were determined as *Ntarget* = 2^ΔCtsample^, where the ΔCt value of the sample was determined by subtracting the average Ct value of the specific target gene from the average Ct value of the *TBP* gene. The *Ntarget* values of the samples were subsequently normalized so that the median of the *Ntarget* values for normal gastric tissues (n = 11) was 1. Target gene expression was normalized to their transcription level of house-keeping genes TATA-Box Binding Protein (TBP), Po, and Peptidylprolyl Isomerase A (PPIA). The preliminary analysis of gene expression did not indicate changes in the median basal levels in normal samples in the same patients (with either diffuse or intestinal GC). For each gene expression, the normalized RNA values of 3 (or more) were considered to represent gene overexpression in tumor samples, and values 0.33 (or less) represented gene under expression.

### 2.3. Immunohistochemistry

Immunohistochemical staining (IHC) was performed on paraffin sections (4 μm) as previously described [12,29]. Immunohistochemical staining for AhR (Santa Cruz Biotechnology, Dallas, TX, USA, H-211, sc-5579, dilution 1/50) and CYP1B1 (Santa Cruz Biotechnology, H-105, sc-32882, dilution 1/200) was performed using the Ventana Autostainer (Export, PA, USA). Specificity was checked by control staining performed in the absence of primary antibodies and with positive tissue [12]. The antigen–antibody complex was visualized using DAB as the chromogen. Immunostaining was analyzed blindly in duplicate by two specialists including a certified pathologist.

### 2.4. Cell Culture

The two human gastric cell lines were obtained from ATCC (Manassas, VA, USA). KATO-III is derived from a poorly differentiated gastric adenocarcinoma, and AGS from moderately differentiated GC were acquired from ATCC (Manassas, VA, USA). The cells were grown in Dulbecco’s modified Eagle medium supplemented with 10% heat-inactivated fetal bovine serum, 0.5% penicillin-streptomycin, and 2 nM of L-glutamine (Gibco, Saint Aubin, France at 37 °C in a humidified 5% CO_2_/95% air atmosphere. Exponentially growing cells were trypsinized and seeded in flasks; the medium was replaced every 24 h; when cells reached 70–80% of confluence as evaluated by microscopic examination, the medium was changed, and either TCDD (1–30 nM) or BaP (10 μM) (gift from P Balaguer, Montpellier, France) was added for 16 h to 24 h. Control experiments included the addition of CHH 223191 (10 μM), a full AhR antagonist (a gift from P Balaguer, Montpellier, France).

### 2.5. Statistical Analysis

As mRNA expression levels did not fit a Gaussian distribution, the relative expression of genes was characterized by the median and the range rather than by their mean values and coefficient of variation [10,12,29]. For each gene, the differences in expression between tumors versus normal tissues (fold change) were analyzed as previously described [10,12]. Differences in the number of samples that over- (>3-fold) or under- (<3-fold) expressed were analyzed using the Chi2-square test [29]. The relationships between the expressions of genes in GC were determined using non parametric Spearman’s rank correlation test. Relationships between the expression levels and clinical parameters were analyzed using non parametric Kruskal–Wallis (or Mann–Whitney) and Chi-square tests, as indicated in each Table. Statistical analyses were performed using the Prism 5.03 software (GraphPad, San Diego, CA, USA). The differences were considered significant at confidence levels greater than 95% (*p* < 0.05).

## 3. Results

### 3.1. Patient Characteristics

The clinical characteristics of the patients are shown in Table 1. The patients with diffuse GC were younger than the patients with intestinal GC (n = 57, 27–71 years and n = 75, 59–82 years, respectively, *p* = 0.0004). Both subtypes of carcinoma had large tumors (>50 mm) and tumor invasion (T3–T4) [10]. Within each subtype, half of the patients smoked. The patients with diffuse adenocarcinoma had more lymphatic invasion (*p* = 0.0014) and metastasis (31% vs. 6%) than the patients with intestinal GC. Most diffuse GC had a TNM stage III–IV, while the patients with intestinal GC were stage I, II, and III. Vascular and neural invasion did not differ among different GC subtypes.

### 3.2. High AhR Expression in Gastric Tumors Both at the mRNA and Protein Levels

The cohort of GC specimens was first used to assess *AhR* mRNA expression levels. As compared to the normal gastric tissue samples, *AhR* expression was significantly increased in gastric tumors (×1.94, *p* = 0.002), both diffuse and intestinal GC (×2.12, *p* = 0.001 and ×1.60, *p* = 0.003, respectively) (Table 2). Moreover, *AhR mRNA* expression was independent of classical clinical parameters, i.e., gender, age, tumor grade, lymphatic invasion, metastasis status, TNM stage, vascular or neural invasion, and the GC subtype (Appendix A).

Most importantly, at the protein expression levels in GC, strong nuclear AhR staining was observed (Figure 1B,D) in epithelial and stromal cells, including fibroblasts and endothelial and immune cells [29], as compared to the weak cytoplasmic and nuclear staining observed in the epithelial cells in non-tumoral tissue (Figure 1A).

Altogether, these results indicate that AhR (mRNA and protein) is significantly increased in GCs, both in diffuse and intestinal adenocarcinoma, as compared to the control samples.

### 3.3. Expression of AhR-Target Genes Encoding Xenobiotic Metabolizing Enzymes in Gastric Cancers

As GC expresses high AhR expression levels (mRNA and protein), we analyzed the changes in the expression of the classic target genes of AhR, such as *CYP1A1* and *CYP1B1*, two genes involved in xenobiotic metabolism [16]. *CYP1A1* was expressed at a low level in normal tissues (Table 2). As compared to the non-tumoral tissue, the enhanced expression of *CYP1A1* was attained with an increase of more than 3-fold observed in 5/29 of GC cases (17%) and in 23% diffuse GC. *CYP1B1* expression was significantly increased in diffuse GC (*p* = 0.014, 92% lymphatic invasion, Table 2 and Table 3). At the protein level, CYP1B1 was mainly observed in the stromal compartment in diffuse GC (Figure 1C). In all the tumors, *CYP1B1* expression was independent of clinical parameters (gender, age, smoking, and tumor grade (T), except for an increase with lymphatic invasion (*p* < 0.02) and TNM (*p* < 0.05 respectively) (Table 3).

### 3.4. AhR Ligands Such as Environmental Ligands Induced mRNA Expression of CYP1A, IL1B, UGT1A1, and AhRR in Gastric Epithelial Cell Lines

Next, it was important to directly evaluate how exposure to POPS impacted the expression of AhR and AhR-related genes. We thus examined how TCDD, the most potent known environmental AhR ligand, and BaP impacted the mRNA expression levels of AhR and AhR-related genes in two epithelial gastric cell lines (KATO III and AGS). These cells are poorly (KATO III) and moderately (AGS) differentiated, with high *AhR* expression.

In the KATO III cells, TCDD (30 nM) strongly increased the expression of *CYP1A1* mRNA levels as compared to untreated cells (×15, *p* < 0.0001) (Figure 2a). The TCDD-induced *CYPA1* expression was reversed by CHH 223191, a full AhR antagonist (*p* < 0.001) (Figure 2a). BaP (10 μM), a well-studied pro-carcinogen, also increased *CYP1A* expression (×5, *p* < 0.001) (Figure 2a). Further, TCDD (30 nM) also significantly increased the expression of *IL1B* (×5.5, *p* < 0.0001), *UDP Glucuronosyltransferase Family 1 Member A complex Locus* (*UGT1A)*, and *AhRR* (×2, *p* < 0.0001) (Figure 2a). We did not detect significant effects of TCDD (or BaP) on *AhR* and *AhRR nuclear translocator (ARNT)* expression in the KATO III cells (Figure 2a). Just as what was seen in the diffuse GC patients, the *CYP1B1* expression levels were increased in the Kato III cells upon TCDD treatment, although to a lesser extent compared to what was seen for *CYP1A1*. (Figure 2b). Of note, the levels of *CYP1B1* were low in the epithelial KATO III cells and undetectable (qPCR threshold > CT50) in the AGS cells. In AGS cells, the expression of *CYP1A* but not *AhR* or *AhRR* was significantly increased following the TCDD treatment (Figure 2c).

### 3.5. Correlations of Expression of AhR, CYP1A1 and CYP1B1, AhRR with a Panel of Genes Involved in AhR-Related Signaling Pathways

AhR is known to activate several signaling pathways governing proliferation, epithelial–mesenchymal transition (EMT), cell migration, inflammation, immunity, and angiogenesis in cancers [12,15,17,18,19,20,21,22,23]. We then compared the mRNA expression levels of *AhR*, *CYP1A1*, and *CYP1B1* with the expression of 36 genes involved in EMT, cell proliferation and migration, immunity, and angiogenesis that we previously described in our cohort of GCs [10,29]. *AhR* expression was correlated with *Ras Homolog Family MemberA* (*RHOA)* expression in intestinal and diffuse GC (Table 4). In diffuse GC, *CYP1A1* expression correlated with several genes such as growth factors (Insulin-Like Growth Factor 1 (*IGF1)*, *p* = 0.001, and IGF Receptor *(IGFR2)*, *p* = 0.015); genes involved in EMT such as *Vimentin (VIM) (p* = 0.007), *Snail Family Transcriptional Repressor* (*SNAI2)* and *Zinc Finger E-Box Binding Homeobx 2* (*ZEB2)* (*p* = 0.04); and migration (*Matrix Metallopeptidase 2* (*MMP)2*, *p* = 0.01) (Table 4). *CYP1B1* expression was only inversely correlated with *Erb-B2 Receptor Tyrosine Kinase 2* (*ERRB2) (p* = 0.01). Moreover, significantly increased *AhRR* expression (×2.65, *p* < 0.01) positively correlated with *IGF1 (p* = 0.0001), *Twist Family BHLH Transcription Factor 2 (TWIST2)*, *ZEB2 (p* < 0.02), *MMP2* (*p* < 0.002), and *Neuropilin 1* (*NRP1)* (*p* = 0.02) in diffuse, but not intestinal GC (Table 4). In intestinal GC, the low expression of *CYP1A1* and *CYPB1* inversely correlated with *Indoleamine 2,3-Dioxygenase 1* (*IDO1)* (*p* = 0.014 and *p* = 0.001, respectively, Table 4). Low *CYP1B1* expression also correlated with genes that were not significantly increased including growth factors and receptors, EMT (*VIM*, *SNAI2*, and *TWIST2*), *VEGF*, and *NRP1* (Table 4).

## 4. Discussion

Although AhR is well reported for its role in environmental chemical toxicity, as a mediator of the unintended consequences of human pollution, and its involvement in tumor initiation and progression [24,25,26,28,35], the relationship between AhR expression, pollution-linked AhR-dependent function, and Western patients with GCs remains unexplored. Because diffuse GC is increasing in prevalence in Western countries, usually diagnosed at advanced stages, and has no efficacious treatment options, the exploration of its cellular and molecular causes is crucial. We reported significantly high expression of *AhR* in our Western cohort of GC independently of their clinical subtypes [29]. A link between exposure to POPs and diffuse GC is supported by the study of accidentally exposed populations in Seveso [11]. In the study presented here, we have analyzed the expression of AhR and several AhR-regulated genes in a series of gastric tumors including diffuse and intestinal GC. Furthermore, we have studied the impact of two AhR ligands well known for their critical role in cancer development linked to pollution on AhR and AhR-related gene expression [26,27].

Expression of *AhR* and *RHOA*: Using RT-PCR, we found a correlation between the expression of *AhR* and *RHOA* in GCs independently of their subtypes (Table 4), as previously documented for other types of cancers [12,36,37]. A higher expression of *RHOA* has been found in diffuse GC with 85% overexpression (>3 fold) as compared to normal samples, along with 50% in intestinal subtype) [10]. The functional and coordinated role of RHOA in the development of cancers involves several processes such as cell proliferation, migration, invasion, and angiogenesis [36,37,38,39,40]. Increased RHOA activity is correlated with worse overall survival in diffuse patients [41]. Interestingly, *RHOA* transcription has recently been shown to be initiated by a ligand-AhR-ARNT complex. Somatic alterations in *RHOA* and *CDH1* have been reported in aggressive diffuse GC and are generally associated with familial disease [4,42]. However, our Western cohort of diffuse GC as defined by the Lauren classification did not include familial GC.

Expression and distribution of *CYP1A1* and *CYP1B1*: Cytochrome P450-1 enzymes are inducible forms of the cytochrome P450 family of xenobiotic metabolizing enzymes [43,44]. In our study, CYP1A1 was detected at a very low level in the stomach, but overexpressed (>3 fold) in 23% of diffuse GC as compared to non-tumoral tissue. *CYP1A1* was also highly induced by TCDD, a non-genotoxic AhR ligand, in undifferentiated diffuse GC (Kato III) cells. CYP1B1 was the most significantly expressed form in diffuse GC, as previously reported in a wide range of human cancers including breast, colon, lung, and others [43], mainly in the cytoplasm in GC which is expected for an enzyme involved in xenobiotic metabolism. CYP1A1 and CYP1B1 have a central role in tumor development and the activation step of pro-carcinogen compounds such as BaP [16,44,45,46,47,48]. BaP is a prototypical PAH found in tobacco and combustion processes such as biomass burning [49,50].

Our in vitro experiments using two GC cell lines (KATO III and AGS) indicated that TCDD, and to a lesser extent BaP, strongly induced *CYP1A1, UGT1A1*, and *NQO1* in KATO III epithelial cells. The expression of *CYP1B1* was also induced upon TCDD treatment, even though at a much lower level as compared to *CYP1A1. CYP1A1* was also increased following TCDD treatment in AGS cells as compared to unexposed cells. Functional DRE enhancer elements have been identified in vitro and in vivo for AhR target genes including *CYP1A1*, *CYP1B1*, and *NQO1,* which encode phase I and II xenobiotic metabolizing enzymes [51]. Our in vivo study also shows that *CYP1B1* expression is significantly increased (*p* = 0.014) in diffuse GC as compared to intestinal GC. At the protein level, CYP1B1 was mainly observed in the stromal compartment in diffuse GC (Figure 1C). It is well known that stromal cells, such as fibroblasts and macrophages, express CYP1B1, but not CYP1A1, in response to TCDD or benzopyrene [52,53]. Taken together, our results suggest a cell-specific distribution of CYP1B1 and CYP1A1 in diffuse GC. Our results also suggest that activated AhR may contribute to the tumor–stroma interaction (through *CYP1A1* and *CYP1B1*) in diffuse GC.

Expression and functional role of AhR, CYP1A1 and CYP1B1: Animal and clinical data provide evidence for the role of AhR in gastric tumorigenesis, implicating the receptor in the regulation of tumor growth, EMT, migration, invasion, and cancer aggression [24,25,28,35]. We observed a significant increase in *AhR* mRNA expression levels in GC independently of intestinal or diffuse GC subtypes and clinical parameters. In addition, strong nuclear AhR is observed in GC tumors (Figure 1B,D), a subcellular distribution for a transcription factor indicative of its constitutive activation in GC. Altogether, these observations strongly suggest the activation of AhR in GC, as previously reported in breast cancer [12]. The connection between EMT and tumorigenesis has been established in human cancers involving several pathways such as the activation of Wnt/β-catenin signaling through CYP1s [54,55], or hedgehog signaling [56,57]. It is also well known that CYP1A1 and CYP1B1 have important roles in tumor development (cell invasion, migration, and disease progression), in part linked to their metabolic activation by BaP. The analysis of CYP1s and co-regulated genes using large-scale analysis may also be helpful for functional studies [58]. Using qRT-PCR, we compared the expression of *CYP1s* with the expression levels of 36 genes coding for proteins that have been previously studied in the same cohort of patients [10,29]. These genes were selected on the basis of their roles in proliferation, the IGF pathway, the EMT signature, migration, angiogenesis, or immunity. In diffuse GC, but not intestinal GC, *CYP1A1* expression was strongly correlated with the expression of genes involved in proliferation (*IGF1*, *p* = 0.001), EMT signature such as *VIM*, *p* = 0.007; *SLUG* and *ZEB2* (*p* = 0.04); and migration (*MMP2*, *p* = 0.01); these genes were previously shown to be correlated with *IGF1* [10]. Interestingly, the promoters of *CYP1A1*, *VIM*, and *SNAI2/SLUG* contain a xenobiotic responsive element (DRE) sequence that when bound by AhR-ARNT heterodimers (canonical pathway) leads to their transcription. The activation of the AhR pathway by TCDD has been previously shown to enhance cancer cell invasion through metalloproteinases [59,60]. Environmental pollutants have been found to contribute to EMT and mesenchymal markers that provide invasion, migration, and subsequent metastasis [60,61,62]. In contrast to diffuse GC, the low expression of *CYP1A1* and *CYP1B1* in intestinal GC inversely correlated with *IDO1* expression (*p* < 0.02). The IDO enzyme mediates the early steps of tryptophan metabolism leading to kynurenine, an endogenous AhR ligand produced in the intestinal but not diffuse GC [29]. Thus, environmental pollutants vs. endogenous kynurenin may have different effects on AhR-dependent gene expression in GCs.

Whether the increased expression of *CYP1s* that we observed in diffuse as compared to intestinal GC is due to exposure to a specific or multiple POPs remains to be established [63]. We have previously reported the significant and widespread increase in a substantial set of POPs such as polychlorinated dioxins (PCDDs/PCDFs), PCBs, and polybrominated flame retardants (such as PBDE 209, a carcinogenic intermediate of BaP) in human omental tissue (fat deposits) from French patients with diffuse GC as compared to control biopsies [13]. The co-exposure of TCDD and PBDE 209 was observed in 33% of the omentum from diffuse GC patients. Interestingly, an increased incidence of hepatocellular carcinomas was observed in rodents upon exposure to PBDE 209 as well as increased *CYP1A1* mRNA expression levels in Caco-2 cells [64].

Expression of *AhRR*: Our results reveal that *AhRR* mRNA expression levels are strongly increased in the cohort of diffuse GC as compared to normal tissues of the same anatomical origin (×2.65, *p* = 0.007) (Table 2). The effect of dioxin on *CYP1A1* and *AhRR* expression in the undifferentiated KATO III as compared to the AGS gastric epithelial cells also supports the role of xenobiotic compounds in vivo. The expression of *AhRR* correlated with *CYP1A1* (*p* = 0.007), *IGF1* (*p* < 0.0001), and with genes involved in EMT (*TWIST2* and *ZEB2*) and cell migration (*MMP2*, *p* = 0.002). Our observations in patients with metastatic diffuse GC further indicate a significant decrease in AhRR expression. Our results suggest that AhRR may represent an independent prognosis factor in diffuse GC, as we previously reported for breast cancer [12]. Poor prognosis was previously correlated with a decreased expression of AhRR in GCs from an Asian gastric cohort, but without discrimination between subtypes [65]. Moreover, the loss of AhRR correlates with an aggressive tumorigenic phenotype in several tumors including colon, cervical, and ovarian carcinoma [66].

In conclusion, this pilot study explores two forms of GCs, diffuse and intestinal, which lead to metastases in the peritoneal cavity. The induction of CYP1s through AhR activation may potentially serve as a biomarker for exposure to xenobiotics in diffuse GC. In vitro experiments indicate that TCDD strongly induces *CYP1A1* in epithelial cells. The expression of *CYP1A1* strongly correlated with genes involved in EMT and migration, and the IGF pathway. The increased expression of CYP1B1 was observed in diffuse GC. CYP1B1 activates a large number of pollutants which may result in the activation of pro-cancer signaling pathways. AhR may contribute to the tumor–stroma interaction (through *CYP1A1* and *CYP1B1)* in diffuse GC. Whether clinical factors such as smoking are prognostic factors remains to be investigated in GC. We argue that reduction in exposure to subsets of environmental ligands could be important to prevent primary diffuse GC. A recent study revealed that exposure to environmental pollutants such as POPs and BaP may reduce the efficacy of chemotherapy [67].

We acknowledge that our study has limitations. Because of the relatively low sample size in this report (n = 29), certainly, the results need to be confirmed using a larger cohort of gastric tumor samples with different clinical characteristics (including early and advanced stages). Nonetheless, our pilot study shed light on the impact of AhR and related signaling pathways in Western patients with GC. In addition, it will be interesting to extend the coverage to different geographical population settings. This will allow us to understand if the signaling pathways identified in GC subtypes are characteristic of Western patients or can also be observed in patients from other geographical locations. Further in vitro and in vivo studies with a larger cohort of gastric tumor samples will provide a better understanding of the complexity of the effect of different ligands on the regulation of the AhR pathways and may contribute to the development of novel clinically relevant agonists or antagonists.

As summarized in Figure 3, the present study provides new insights into the diversity of AhR functions in the development of cancer including GC. It is likely that the binding of various ligands is central to this carcinogenesis. The gastric epithelium is constantly exposed to exogenous AhR ligands such as dietary compounds and environmental toxins (PAH and other dioxin-like compounds), which enable the strong activation of AhR. Furthermore, the endogenous AhR kynurenine is produced through the metabolism of tryptophan by IDO1 which is induced in stromal cells, or by TDO2 which can be up-regulated in tumor cells and the tumor stroma. Our findings merit further studies with a larger cohort of gastric tumor samples with different clinical characteristics, including early and advanced tumor stages.

## Figures and Tables

**Figure 1 biomedicines-12-01905-f001:**
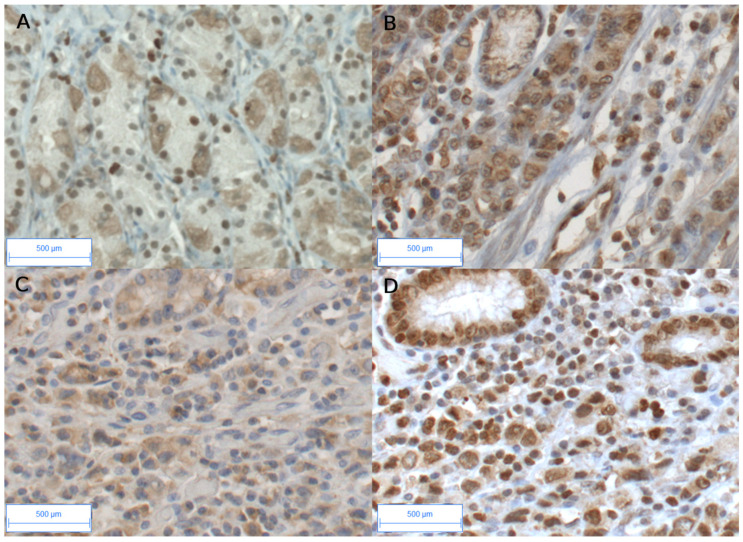
Immunohistochemical staining of AhR and CYP1B1 in peritumoral and diffuse GCs. AhR in peritumoral gastric tissue (**A**); weak cytoplasmic and/or nuclear staining were observed in glandular tissue and stroma. In tumoral tissue (**B**,**D**), strong AhR immunostaining is observed in most cells, both epithelial and stromal compartments. CYP1B1 (**C**) and AhR (**D**) immunostaining are shown on the same tumor (diffuse GC). CYP1B1 was mainly observed in the stromal compartment in diffuse GC (**C**). Original magnification ×20. Bar scale, 500 μm.

**Figure 2 biomedicines-12-01905-f002:**
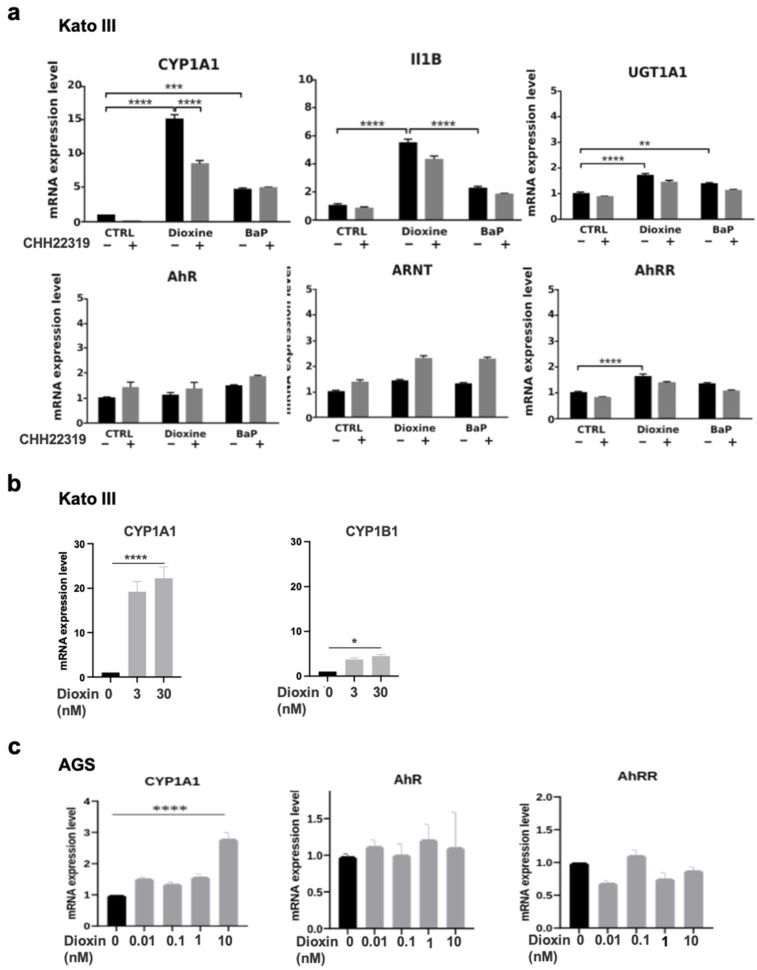
mRNA expression levels of AhR and AhR-related genes in KATO III and AGS gastric cells upon treatment with either TCDD or BaP. (**a**) The KATO III cells were cultivated in the absence (Ctrl) or presence of either TCDD (dioxin) 30 nM or BaP (10 µM) for 16 h. The cells were incubated with (gray column) or without (black column) CHH223191 (10 μM). The expression of the indicated genes was determined by qRT-PCR. All the experiments were performed in triplicate. The results are expressed as means +/− S.E.M and normalized so that the mean of the control cells was 1. * *p* value < 0.005, ** *p* value < 0.01; *** *p* value < 0.001; **** *p* value < 0.0001. (**b**) The KATO III cells were cultivated in the presence or absence of dioxin at the indicated concentrations. The expression levels of *CYP1A1* and *CYP1B1* were determined by qRT-PCR in the same experiment. The results were expressed as in (**a**). (**c**) The AGS cells were cultivated in the absence (Ctrl in black) or presence of (dioxin) (0.01–10 nM, in gray). The expression levels of the indicated genes were determined by qRT-PCR in the same experiment. All the experiments were performed in triplicate. The results were expressed as in (**a**).

**Figure 3 biomedicines-12-01905-f003:**
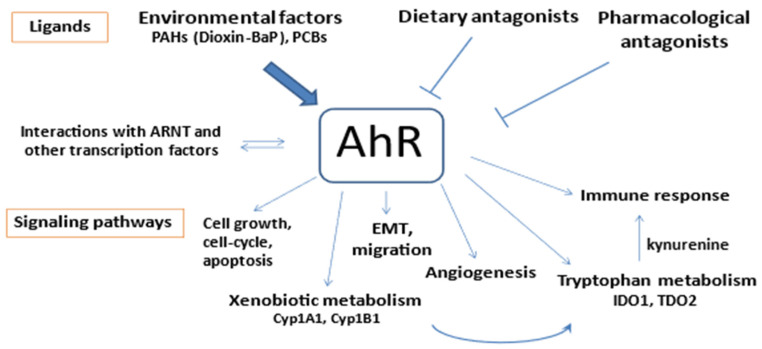
AhR role in cancer biology; environmental compounds at the crossroads of toxicity and several signaling pathways.

**Table 1 biomedicines-12-01905-t001:** Clinicopathological characteristics of gastric carcinoma patients accordingly with histopathological subtype.

	Total GC (n = 29)	Diffuse/Poorly Cohesive GC * (n = 13) (45%)	Intestinal-Subtype GC ** (n = 16) (55%)	*p* Value ^a^
Gender, n (%)				
Male	13/29	6/13 (46%)	7/16 (43%)	0.90 (NS)
Female	16/29	7/13 (54%)	9/16 (56%)	
Age (years, median)	63 ± 17	57 (27–71)	75 (59–82)	0.0004 ^b^
Smoking				
Negative	12/22	4/12	8/12	0.77 (NS)
Positive	10/22	3/10	7/10	
Tumor size (mm), n (%)				
<50	10/27	4/11 (36%)	6/16 (37%)	0.10 (NS) ^b^
≥50	17/27	7/11 (64%)	10/16 (63%)	0.95 (NS)
Depth tumor invasion (T)				
T1–T2	6/29	2/13 (15%)	4/16 (33%)	0.5 (NS)
T3–T4	23/29	11/13 (85%)	12/16 (67%)	
Lymphatic invasion (N)				
Negative	11/28	1/13 (7%)	10/15 (67%)	0.0014
Positive	17/28	12/13 (92%)	5/15 (33%)	
Metastasis (M), n (%)				
Negative	24/29	9/13 (69%)	15/16 (94%)	0.14 (NS)
Positive	5/29	4/13 (31%)	1/16 (6%)	
TNM status				
I–II	16/29	5/13 (38.5%)	11/16 (69%)	0.10 (NS)
III–IV	13/29	8/13 (61.5%)	5/16 (31%)	
Vascular invasion, n + (%)				
Negative	9/29	3/13 (23%)	6/16 (38%)	0.67 (NS)
Positive	20/29	10/13 (77%)	10/16 (62%)	
Neural invasion, n (%)				
Negative	23/29	2/13 (15%)	4/16 (25%)	0.66 (NS)
Positive	6/29	11/13 (68%)	12/16 (75%)	

* poorly cohesive adenocarcinoma/diffuse-type carcinoma; ** intestinal-type adenocarcinoma. a, Chi-square test, Yates’ continuity corrected chi-square test, or Fisher’s exact test if appropriate; b, Mann–Whitney. NS = not statistically different. The characteristics of the patients included in the cohort were already reported in ref. [29].

**Table 2 biomedicines-12-01905-t002:** Statistical analysis of mRNA expression levels of *AhR*, *CYP1B1*, *CYP1A1*, and *AhRR* in gastric cancers (all GC, diffuse and intestinal subtypes).

Genes	Nontumoral Gastric Tissues (n = 11)	All Tumors (n = 29)	*p* Value ^a^	Diffuse GC vs. PT (n = 13)	*p* Value ^a^	Intestinal GC vs. PT (n = 16)	*p* Value ^a^	*p* Value (GC Subtypes)
**AhR and Target Genes**								
** *AhR* **	1 (0.37–1.64)	**1.94 (0.55–3.53)**	**0.002**	**2.12 (0.55–3.35)**	**0.001**	**1.60 (0.65–3.53)**	**0.003**	0.13 (NS)
** *CYP1B1* **	1 (0.52–2.90)	1.45 (0.13–4.90)	0.91 (NS)	**1.62** (0.43–4.90)	**0.014**	1.22 (0.13–4.0)	0.82 (NS)	0.19 (NS)
** *CYP1A1* **	0 (0–5.6)			1.37 (0–86)	0.26 (NS)	0.43 (0–30)	0.73 (NS)	NS
** *AhRR* **	1 (0.23–1.66)	1.25 (0.19–3.93)	>0.999 (NS)	2.65 (0.74–3.96)	**0.007**	0.89 (0.19–3.85)	0.88 (NS)	**0.017**

Median (range) of gene mRNA expression levels; *p* value (^a^ Mann–Whitney). Significant *p* value in bold. Comparative basal levels of genes in normal gastric tissue (×1) are as follows: *AhR* (71), *CYP1A1* (0), *CYP1B1* (372), and *AhRR* (21). NS = not statistically different. *AhR* expression was already reported in ref. [29].

**Table 3 biomedicines-12-01905-t003:** Relationship between *CYP1B1*, *CYP1A1*, and *AhRR* expression with clinical parameters in gastric adenocarcinomas (including sub-populations in all tumors (**A**) and diffuse and intestinal GC subtypes (**B**).

**(A)**								
		**All tumors, n = 29**						
		** *AhRR* **		** *CYP1B1* **			** *CYP1A1* **	
Gender		*p* = 0.75		*p* = 0.47			*p* = 0.063	
Male (n = 13)		1.63 (0.29–3.96)		1.62 (0.13–4.03)			*1.37 (0–30.1)*	
Female (n = 16)		1.18 (0.29–3.85)		1.34 (0.32–4.9)			*0 (0–86)*	
Age		*p* = 0.27		*p* = 0.18			*p* = 0.97	
<60 years (n = 9)		1.66 (0.74–3.96)		1.47 (1.23–4.9)			0.67 (0–30.1)	
>60 years(n = 20)		1.17 (0.19–3.85)		1.21 (0.13–4)			0.65 (0–86)	
Smoking		*p* = 0.54		*p* = 0.34			*p* = 0.42	
Negative (n = 12)		1.21 (0.42–1.67)		1.63 (0.36–4)			0.33 (0–1.98)	
Positive (n = 10)		1.65 (0.29–3.85)		0.98 (0.13–5)			0.43 (0–86)	
Tumor invasion (T)		*p* = 0.11		*p* = 0.18			*p* = 0.63	
T1–T2 (n = 6)		0.43 (0.19–3.5)		0.94 (0.13–2.44)			0.47 (0–86)	
T3–T4 (n = 23)		1.31 (0.42–3.87)		1.47 (0.32–4.9)			0.67 (0–30.1)	
Lymphatic invasion		*p* = 0.032		*p* = 0.014			*p* = 0.06	
Negative (n = 11)		0.74 (0.19–3.85)		0.5 (0.3–2.5)			*0 (0–1.39)*	
Positive (n = 17)		1.67 (0.42–3.96)		1.6 (0.4–4.9)			*1 (0–86)*	
Metastasis (M)		*p* = 0.59		*p* = 0.55			*p* = 0.59	
Negative (n = 24)		1.28 (0.19–4.0)		1.39 (0.13–4)			0.68 (0–86)	
Positive (n = 5)		1.15 (0.74–2.38)		1.45 (0.43–4.9)			0 (0–30.1)	
TNM		*p* = 0.35		*p* = 0.045			*p* = 0.10	
I–II (n = 16)		1.06 (0.19–3.96)		1.19 (0.13–3)			0.12 (0–4.01)	
III–IV (n = 13)		1.63 (0.42–3.66)		1.62 (0.43–4.9)			1.37 (0–86)	
Vascular invasion,		*p* = 0.09		*p* = 0.39			*p* = 0.45	
Negative (n = 9)		0.74 (0.2–4)		1.27 (0.36–2.32)			0.61 (0–4.01)	
Positive (n = 20)		1.65 (0.3–3.8)		1.54 (0.13–4.9)			0.71 (0–86)	
Neural invasion		*p* = 0.38		*p* = 0.41			*p* = 0.38	
Negative (n = 6)		0.79 (0.19–3.53)		1.76 (0.72–2.51)			1.04 (0–86)	
Positive (n = 23)		1.31 (0.29–3.96)		1.4 (0.13–4.9)			0.61 (0–30.1)	
**(B)**								
	**Diffuse subtype GC, n = 13**				**Intestinal subtype GC, n = 16**			
	**n=**	** *AhRR* **	** *CYP1B1* **	** *CYP1A1* **	** *n* ** **=**	** *AhRR* **	** *CYP1B1* **	** *CYP1A1* **
Gender		*p* = 0.42	*p* = 0.92	*p* = 0.83		*p* = 0.88	*p* = 0.67	*p* = 0.058
Male	6	2.2 (1.23–3.96)	1.76 (1.15–2.32)	1.38 (0–4)	7	0.93 (0.29–2.38)	1.45 (0.13–4)	*0.76 (0–30.1)*
Female	7	2.65 (0.74–3.5)	1.47 (0.43–4.9)	0 (0–86)	9	0.85 (0.19–3.85)	1.16 (0.32–2.5)	*0 (0–1.39)*
Age		*p* = 0.27	*p* = 0.49	*p* = 0.16		ND	ND	ND
<60 years	8	1.64 (0.74–3.96)	1.68 (1.23–4.9)	0.33 (0–4)	1	2.38	1.45	30.1
>60 years	5	3.17 (1.15–3.66)	1.62 (0.43–2.44)	1.51 (0–86)	15	0.85 (0.19–3.85)	1.16 (0.13–4)	0.25 (0–3.31)
Smoking		*p* = 0.46	*p* = 0.23	*p* = 0.40		*p* = 0.84	*p* = 0.054	*p* = 0.86
Negative	4	1.43 (0.74–1.66)	1.78 (1.31–2.32)	0.33 (0–1.37)	8	1.02 (0.42–1.67)	*1.63 (0.36–4)*	0.38 (0–1.98)
Positive	3	3.17 (0.74–3.53)	2.44 (1.85–4.9)	62.55 (0–86)	7	0.93 (0.29–3.85)	*0.49 (0.13–3)*	0.25 (0–30.1)
Tumor invasion		ND	ND	ND		*p* = 0.002	*p* = 0.055	*p* = 0.36
T1–T2	2	3.35 (3.17–3.53)	2.14 (1.85–2.44)	74.3 (62.5–86)	4	0.31 (0.19–0.53)	*0.46 (0.13–1.2)*	0.12 (0–0.7)
T3–T4	11	1.66 (0.74–3.96)	1.47 (0.43–4.9)	0.67 (0–4)	12	1.22 (0.42–3.85)	*1.52 (0.32–4)*	0.68 (0–30.1)
Lymphatic invasion		ND	ND	ND		*p* = 0.25	*p* = 0.019	*p* = 0.04
Negative	1	1.63	2.32	1.37	10	0.73 (0.19–3.85)	0.52 (0.13–2.5)	0 (0–1.39)
Positive	12	2.7 (0.74–3.96)	1.54 (0.43–4.9)	1.03 (0–86)	5	1.31 (0.42–2.38)	1.59 (1.27–4)	1 (0–30.1)
Metastasis		*p* = 0.006	*p* = 0.82	*p* = 0.06		ND	ND	ND
Negative	9	3.17 (1.23–3.96)	1.62 (1.15–2.44)	*1.51 (0–86)*	15	0.85 (0.19–3.85)	1.16 (0.13–4)	0.25 (0–3.31)
Positive	4	0.94 (0.74–1.63)	1.86 (0.43–4.9)	*0 (0–1.37)*	1	2.38	1.45	30.1
TNM		*p* = 0.50	*p* = 0.68	*p* = 0.83		*p* = 0.22	*p* = 0.038	*p* = 0.097
I–II	5	2.65 (1.23–3.96)	1.47 (1.23–2.17)	0.67 (0–4)	11	0.74 (0.19–3.85)	0.52 (0.13–3)	0 (0–3.31)
III–IV	8	2.19 (0.74–3.66)	1.73 (0.43–4.9)	1.38 (0–86)	5	1.31 (0.42–2.38)	1.59 (1.27–4)	1 (0–30.1)
Vascular invasion		*p* = 0.84	*p* = 0.57	*p* > 0.9999		*p* = 0.17	*p* = 0.53	*p* = 0.41
Negative	3	1.63 (0.74–3.96)	1.9 (1.4–2.32)	1.37 (0–4)	6	0.62 (0.19–1.31)	0.94 (0.36–1.6)	0.30 (0–1)
Positive	10	2.70 (0.74–3.66)	1.54 (0.43–4.9)	1.03 (0–86)	10	1.09 (0.29–3.85)	1.56 (0.13–4)	0.5 (0–30.1)
Neural invasion		ND	ND	ND		*p* = 0.12	*p* = 0.52	*p* = 0.76
Negative	2	3.35 (3.17–3.53)	2.14 (1.85–2.44)	74.3 (62.5–86)	4	0.63 (0.19–0.85)	1.41 (0.72–2.5)	0.35 (0–1.39)
Positive	11	1.66 (0.74–3.96)	1.47 (0.43–4.9)	0.67 (0–4)	12	1.22 (0.29–3.85)	0.89 (0.13–4)	0.43 (0–30.1)

Median (range) of gene mRNA expression levels; *p* value (Mann–Whitney). Significant *p* value in bold; tendency in italic. ND = not determined.

**Table 4 biomedicines-12-01905-t004:** Correlations of the selected genes analyzed in the study in relation to AhR and “AhR-related signalling pathways” in diffuse (**A**) and intestinal (**B**) GC.

**(A)**								
	**Diffuse GC**							
**Genes**	** *AhR* **	** *AhR* **	** *AhRR* **	** *AhRR* **	** *CYP1B1* **	** *CYP1B1* **	** *CYP1A1* **	** *CYP1A1* **
	**r**	***p* value**	**r**	***p* value**	**r**	***p* value**	**r**	***p* value**
*AhR*	x		−0.031	0.92	−0.066	0.83	0.181	0.553
*AhRR*	−0.031	0.92	x		−0.107	0.73	0.711	0.007
*CYP1B1*	−0.066	0.83	−0.107	0.73	x		0.342	0.253
Growth factors and receptors (n = 10)								
*IGF1*	0.184	0.55	0.865	0.0001	0.155	0.71	0.846	0.001
*IGF1R*	0.596	0.032	0.267	0.38	−0.159	0.6	0.345	0.24
*FGFR1*	−0.234	0.441	0.542	0.055	0.269	0.37	0.444	0.13
*FGF7*	0	1	0.576	0.04	−0.055	0.86	0.424	0.15
*IGF2*	0.041	0.89	0.119	0.7	−0.033	0.91	−0.043	0.69
*IGFR2*	0.562	0.046	0.375	0.21	−0.06	0.84	0.655	0.015
*IRS1*	−0.259	0.39	0.457	0.12	−0.121	0.69	0.384	0.19
*IRS2*	0.259	0.39	0.501	0.08	0.005	0.99	0.623	0.02
*ERBB2*	0.341	0.25	0.529	0.06	−0.676	0.01	0.291	0.34
EMT and migration (n = 10)								
*VIM*	0.135	0.66	0.518	0.07	0.280	0.35	0.709	0.007
*CDH1*	0.501	0.08	0.57	0.09	−0.264	0.38	0.592	0.03
*SNAI1*	0.239	0.43	0.102	0.74	0.066	0.83	0.131	0.67
*TGFB1*	0.297	0.32	0.182	0.55	−0.511	0.83	−0.165	0.59
*RUNX3*	−0.771	0.8	0.202	0.51	−0.313	0.30	−0.065	0.83
*SNAI2*	0.317	0.29	0.441	0.13	0.044	0.89	0.605	0.04
*TWIST2*	−0.005	0.99	0.667	0.013	0.115	0.71	0.504	0.08
*ZEB2*	0.033	0.91	0.661	0.014	−0.038	0.90	0.602	0.04
*RHOA*	0.600	0.034	0.176	0.56	−0.203	0.50	0.484	0.09
*RHOB*	−0.215	0.48	0.295	0.23	0.429	0.14	0.444	0.13
Cell proliferation and migration (n = 3)								
*Ki67*	0.463	0.11	0.328	0.27	−0.176	0.56	0.537	0.26
*MMP2*	−0.099	0.74	0.774	0.002	0.164	0.59	0.701	0.01
*MMP9*	0.193	0.53	0.005	0.99	−0.511	0.07	−0.271	0.32
Immunity (n = 5)								
*IDO1*	0.528	0.07	−0.228	0.44	−0.440	0.13	−0.245	0.42
*TDO2*	0.267	0.38	0.330	0.27	−0.280	0.35	0.048	0.87
*PD1*	0.534	0.06	0.033	0.92	0.115	0.71	0.209	0.49
*CD274*	0.446	0.13	−0.437	0.13	−0.060	0.85	−0.435	0.11
*PDL2*	0.332	0.26	0.080	0.79	0.091	0.77	0.266	0.38
Angiogenesis (n = 6)								
*FLT1*	0.559	0.047	0.303	0.31	0.214	0.48	0.319	0.29
*VEGF165*	0.402	0.17	0.358	0.23	0.011	0.97	0.364	0.22
*VEGF189*	0.306	0.31	0.088	0.77	−0.137	0.65	0.114	0.71
*KDR*	−0.187	0.54	−0.151	0.62	−0.368	0.22	−0.387	0.19
*VEGFC*	0.179	0.56	0.328	0.27	0	1	0.199	0.51
NRP1	0.185	0.55	0.614	0.02	0.033	0.91	0.411	0.16
**(B)**								
	**Intestinal GC**							
**Genes**	** *AhR* **	** *AhR* **	** *AhRR* **	** *AhRR* **	** *CYP1B1* **	** *CYP1B1* **	** *CYP1A1* **	** *CYP1A1* **
	**r**	***p* value**	**r**	***p* value**	**r**	***p* value**	**r**	***p* value**
*AhR*	1		0.679	0.004	0.018	0.95	0.112	0.68
*AhRR*	0.679	0.004	1		0.253	0.34	0.241	0.368
*CYP1B1*	0.018	0.95	0.253	0.34	1		0.655	0.006
Growth factors and receptors (n = 10)								
*IGF1*	−0.113	0.68	0.168	0.53	0.765	0.001	0.371	0.16
*IGF1R*	−0.053	0.84	0.077	0.78	0.667	0.005	0.521	0.04
*FGFR1*	−0.021	0.94	0.132	0.62	0.794	<0.0001	0.455	0.08
*FGF7*	−0.127	0.64	0.047	0.86	0.721	0.002	0.401	0.12
*IGF2*	−0.093	0.73	0.144	0.59	0.774	<0.0001	0.337	0.2
*IGFR2*	0.169	0.53	0.018	0.95	0.524	0.037	0.364	0.15
*IRS1*	0.056	0.84	0.471	0.07	0.915	<0.0001	0.531	0.05
*IRS2*	0.113	0.68	0.203	0.45	0.582	0.018	0.05	0.83
*ERBB2*	0.533	0.03	0.288	0.28	−0.141	0.602	0.097	0.72
EMT and migration (n = 10)								
*VIM*	0.186	0.49	0.132	0.62	0.812	<0.0001	0.379	0.15
*CDH1*	0.284	0.286	0.085	0.75	0.051	0.85	0.214	0.43
*SNAI1*	0.087	0.75	0.068	0.8	0.577	0.02	0.241	0.37
*TGFB1*	0.21	0.39	−0.091	0.74	0.453	0.08	0.118	0.66
*RUNX3*	0.195	0.47	0.081	0.76	−0.199	0.46	−0.071	0.8
*SNAI2*	0.282	0.29	0.171	0.53	0.711	0.002	0.131	0.63
*TWIST2*	0.121	0.66	0.109	0.69	0.827	<0.0001	0.307	0.25
*ZEB2*	0.133	0.62	0.242	0.37	0.477	0.001	0.462	0.07
*RHOA*	0.693	0.003	0.324	0.22	−0.056	0.84	0.171	0.53
*RHOB*	−0.094	0.73	0.041	0.88	0.559	0.02	0.201	0.45
Cell proliferation and migration (n = 3)								
*Ki67*	0.277	0.3	−0.135	0.61	−0.665	0.006	−0.381	0.1
*MMP2*	0.121	0.65	0.118	0.66	0.800	<0.0001	0.335	0.2
*MMP9*	0.139	0.61	0.041	0.88	0.185	0.49	0.07	0.8
Immunity (n = 5)								
*IDO1*	0.272	0.3	−0.025	0.94	−0.727	0.001	−0.601	0.014
*TDO2*	0.046	0.87	0.103	0.7	0.218	0.41	0.163	0.54
*PD1*	−0.155	0.56	−0.132	0.62	0.056	0.84	−0.261	0.25
*CD274*	0.199	0.46	0.156	0.56	−0.251	0.34	−0.332	0.15
*PDL2*	−0.015	0.95	−0.172	0.517	0.455	0.08	−0.032	0.76
Angiogenesis (n = 6)								
*FLT1*	−0.407	0.12	−0.394	0.13	0.032	0.91	−0.176	0.51
*VEGF165*	−0.531	0.03	−0.641	0.01	−0.321	0.23	−0.013	0.68
*VEGF189*	−0.181	0.51	−0.238	0.37	−0.156	0.56	−0.046	0.86
*KDR*	−0.081	0.77	−0.079	0.77	0.482	0.06	0.061	0.82
*VEGFC*	0.01	0.88	0.132	0.63	0.788	0.0001	0.353	0.18
*NRP1*	0.131	0.62	0.168	0.53	0.838	<0.0001	0.324	0.22

r, Spearman’s rank test (relationship between two quantitative parameters). Values in bold type are statistically significant at confidence level greater than 99% (*p* value < 0.01) and r > 0.6. *CYP1B1* and *AhRR* were not significantly increased in intestinal GC and statistical correlations should be considered with caution (see Table 2). x, maximum correlation = 1.

## Data Availability

A total of 29 patients underwent partial gastrectomy for histopathologically confirmed gastric adenocarcinoma primary tissue in the Lariboisiere Hospital (Paris, France) from 2005 to 2014.

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
