# Peer review of "High Expression of AhR and Environmental Pollution as AhR-Linked Ligands Impact on Oncogenic Signaling Pathways in Western Patients with Gastric Cancer—A Pilot Study"

_biomedicines, 2024, doi:10.3390/biomedicines12081905_

Round 1
Reviewer 1 Report
Comments and Suggestions for Authors
The authors investigated gene expression of aryl hydrocarbon receptor (AhR) and its related genes in diffuse gastric cancers. They examined 13 diffuse gastric cancers and 16 intestinal-subtype gastric cancers. AhR and downstream gene expression were analyzed using mRNA from fresh-frozen samples of tumors and paired normal tissues, and correlation with other gene expression was also examined. In addition, the expression of AhR and Cyp1B1 proteins in diffuse-type gastric cancer was shown using immunohistochemistry. They showed that AhR gene expression was significantly elevated in both diffuse-type and intestinal-type gastric cancer compared with normal tissue, with expression being particularly elevated in diffuse-type gastric cancer. Using AhR ligands, increased expression of downstream genes such as CYP1A1 was also demonstrated in cultured cells of diffuse gastric cancer. In relation to the expression of other genes, a correlation with RhoA has been shown, and they suggest a correlation with genes related to cell proliferation and migration. In this study, they showed a correlation between AhR gene expression in diffuse gastric cancer, as well as the expression of its downstream genes and genes related to tumor proliferation, epithelial mesenchymal transition (EMT) and invasion. The study may provide an important contribution to carcinogenesis of diffuse-type gastric cancer. The following major issues, however, need authors' attention.
1. The authors also investigated changes in AhR-related gene expression in cultured cells, but failed to show how these cells behaved. They should examine and show proliferation, migration and invasion assay using the cultured cells.
2. They showed that AhR and Cyp1B1 gene expression was high in diffuse-type gastric cancer using clinical specimens. The results of Cyp1B1 gene expression should also be shown in Figure 2.
3. Using immunohistochemistry, they also demonstrated expression of AhR and Cyp1B1 proteins, with positive images seen in the nucleus and cytoplasm. They should provide an explanation for AhR and Cyp1B1, including not only their function but also their intracellular localization.
4. The epidemiological information on gastric cancer in the introduction is outdated, including the WHO classification. They should cite the latest data.
5. The clinicopathological characteristics in Table 1 seems almost identical to those in their previous report (Biomedicines 2022, 10(2), 240). If they used the same cohort as in their previous report, they should state this in the methods and table.
6. If they used the same cohort as their previous reports, this should be confirmed as the number of cases of depth tumor invasion differs from previous reports.
7. The AhR values in Table 2 are the same as those in previous reports, so the citation should be clearly stated within the table.
Author Response
English editing
Following the recommendations, the article has now been deeply corrected by an English speaking person. We hope you will appreciate the great deal of work that was made to improve the revised version (compare the annotated vs the final version) and that the revised version now reaches your expectations.
Here is our point-by-point response to the reviewers’ comments:
Reviewer #1
We thank the referee for the constructive criticisms and suggestions. The following are our answers to the specific points.
- The authors also investigated changes in AhR-related gene expression in cultured cells, but failed to show how these cells behaved. They should examine and show proliferation, migration and invasion assay using the cultured cells.
We thank the reviewer for his/her suggestion. However, we would like to point out that it is already reported that the AhR agonist TCDD (dioxin) promotes GC cell migration and invasion (Peng T-L, BMC Biology, 2009) and GC cell proliferation (Peng, T-L, 2009, World J of Gastroanterology).
Further, our study mainly focuses on changes of mRNA expression levels prepared from fresh-frozen samples of tumors compared to paired normal tissues. Thus, even though, we used Kato III and AGS cell lines as models to study how POPs impact on AhR and AhR-related gene mRNA expression, a 2D-study in such human cancer cell lines isolated decades ago and cultured in various conditions is far from reflecting what could have happened in GC patients after long-term exposure to POPs.
- They showed that AhR and Cyp1B1 gene expression was high in diffuse-type gastric cancer using clinical specimens. The results of Cyp1B1 gene expression should also be shown in Figure 2.
Following the reviewer’s recommendation, for more clarity, we have now merged Figure 2 and Figure 3. CYP1B1 mRNA expression levels are now presented in new Figure 2b.
- Using immunohistochemistry, they also demonstrated expression of AhR and Cyp1B1 proteins, with positive images seen in the nucleus and cytoplasm. They should provide an explanation for AhR and Cyp1B1, including not only their function but also their intracellular localization.
We thank the reviewer for this interesting comment. We have now discussed AhR subcellular expression and included a tentative explanation for the strong AhR nuclear staining in GC in the discussion section, new subheading changed from “Functional role of AhR, CYP1A1 and CYP1B1” to “Expression and functional role of AhR, CYP1A1 and CYP1B1” following the reviewer’s recommendations. As for CYP1B1, a sentence has been added to the discussion section, subheading “Expression and distribution of CYP1A1 and CYP1B1”.
- The epidemiological information on gastric cancer in the introduction is outdated, including the WHO classification. They should cite the latest data.
We thank the reviewer for pointing it out. We have now updated references 1 to 4 in the introduction section.
- The clinicopathological characteristics in Table 1 seems almost identical to those in their previous report (Biomedicines 2022, 10(2), 240). If they used the same cohort as in their previous report, they should state this in the methods and table.
The reviewer is right. Indeed we have used the same cohort as published in Biomedicines, 2022. This is now stated in the methods and Table 1.
- If they used the same cohort as their previous reports, this should be confirmed as the number of cases of depth tumor invasion differs from previous reports.
We thank the reviewer for such a careful reading. While the number of cases of depth tumor invasion is the same as compared to what was previously reported in Biomedicine, 2022, there is a mistake in the p-value that has been corrected to 0.5 (NS) (not 0.66). The correction has been made in Table 1.
- The AhR values in Table 2 are the same as those in previous reports, so the citation should be clearly stated within the table.
We have now added the citation to Table 2.
Reviewer 2 Report
Comments and Suggestions for Authors
This study included 29 paired normal and tumor tissue samples, and the corresponding clinical information, trying to explore the expression of the AhR gene and its potential functions in environmental pollution-involved carcinogenesis of gastric cancer. However, the results only show protein and mRNA levels of AhR, as well as its correlation with Cyp P450 enzymes and pathway-related genes, which are not profound enough to support the conclusions.
1. The manuscript contains numerous grammatical errors, leading to misunderstandings and making it difficult to follow.
2. Figure 1 needs to be quantified (such as index scores and ratios of nuclear/cytoplasmic location) and visualized by bar plot.
3. Table 2 and Table 3: better use bar plot to interpret the results.
4. Figure 2 and Figure 3 can be merged together. The readings of the Y axis in Figure 3A should be normalized to coincide with other panels.
5. Table 4 should also be visualized by plots.
6. The study is not specific to diffuse gastric cancer only. The intestinal subtype was also included. I suggest removing the “Diffuse” word from the title.
Comments on the Quality of English LanguageThe manuscript contains numerous grammatical errors, leading to misunderstandings and making it difficult to follow.
Author Response
English editing
Following the recommendations, the article has now been deeply corrected by an English speaking person. We hope you will appreciate the great deal of work that was made to improve the revised version (compare the annotated vs the final version) and that the revised version now reaches your expectations.
Here is our point-by-point response to the reviewers’ comments:
Reviewer #2
We thank the referee for the constructive criticisms and suggestions. The following are our answers to the specific points.
- The manuscript contains numerous grammatical errors, leading to misunderstandings and making it difficult to follow.
The article has now been deeply corrected by an English speaking person. We hope you will appreciate the great deal of work that was made to improve the revised version (compare the annotated vs the final version) and that the revised version now reaches your expectations.
- Figure 1 needs to be quantified (such as index scores and ratios of nuclear/cytoplasmic location) and visualized by bar plot.
We thank the reviewer for this suggestion. Unfortunately, no pathologist has previously established such a precise scoring and no one is available for further analysis. Thus, no scoring can be established at this stage of the study. We reassure the reviewer that the images included in the manuscript are representative of the increased expression of AhR as observed by the pathologist in the analyzed GC cohort.
- Figure 2 and Figure 3 can be merged together. The readings of the Y axis in Figure 3A should be normalized to coincide with other panels.
The reviewer is right. Following the reviewer’s recommendations, we have now merged Figure 2 and Figure 3. We have also corrected the Y axis of former Figure 3A, that is now included in Figure 2b.
- Table 2 and Table 3: better use bar plot to interpret the results
&
- Table 4 should also be visualized by plots.
We thank the reviewer for his/her appreciation that we have generated a large number of information. Tables allow to group a maximum of information, and further, allow direct comparations between results for optimal interpretation of the data. We would like to point out to the reviewer that a boxplot representation of the data, as an example, included in Table 3, would lead to the generation of 81 boxplots which I am sure he/she will recognize that it is not a way to simplify the interpretation of data. Altogether, this is why the presentation of all these date in Tables is a deliberate choice.
We also would like to point out that many other studies have selected the same optimal way to present similar large set of data, several of them being published in journals from MDPI. To cite some of our previous studies: Perrot-Applanat, M. et al., Biomedicines, 2022; Perrot-Applanat, M. et al., Cancers, 2022, Perrot-Applanat, M. et al., Oncol Lett, 2019; Vacher, S. et al., PLoS One, 2018,….
- The study is not specific to diffuse gastric cancer only. The intestinal subtype was also included. I suggest removing the “Diffuse” word from the title.
Following the reviewer’s recommendations, “diffuse” has been removed from the title.
Comments on the Quality of English Language
The manuscript contains numerous grammatical errors, leading to misunderstandings and making it difficult to follow.
As indicated above (point #1), deep English editing has now been made.
We hope that you and the reviewers will now find this new version of our manuscript to be suitable for publication in Biomedicines
Sincerely,
Martine Perrot-Applanat and Véronique Baud
Reviewer 3 Report
Comments and Suggestions for Authors
The authors really like PCR. This manuscript is mainly driven by gene expression analyses, mostly by PCR, and to some degree, minor validation by immune histochemistry, but not in the cell line used for PCR, but in tissues. Thus, there is a lack of convincing functional data, although many of these are discussed broadly in the last section ("discussion")... but without much proof based on own experiments. Rather, based on publications by others.
Thus the manuscript lacks a bit of genuine novelty and originality, it starts addressing some issues like EMT and tumor cell invasion, but it doesn't even attempt to validate any of this in the only cell line that is used here. Using more than just one cell line, and adding at least SOME functional validation, would also help to bridge the conceptual gap between exploring tumor tissues (in the 1st half of the manuscript) and performing some experiments with a single cell line (Kato III).
This is in my opinion, insufficient to warrant acceptance and publication, and some validation on the cellular level, and NOT based on PCR, should be added.
There are also many smaller issues listed below:
The abstract could be somewhat streamlined, as it's complicated to read with all of the gene names or symbols; few of them are explained as it usually is in an abstract. Maybe the abstract should focus more on the main findings and therefore would also be less complex.
Also, the explanation of abbreviations is sometimes inconsistent in the manuscript, for example, POP is used before it is explained a few lines further down. In contrast, other abbreviations like TCCD and PCBs arent explained at all. Especially, some of the chemicals (like BaP), or chemical classes, are never properly introduced. Sometimes they are explained many pages further down (like TCCD). It would help many of the readers to disclose those names, especially in introduction, at first occurence, and this is also the habit in academic publications.
Generally, abbreviations and gene symbols are sometimes explained, sometimes not. Its just inconsistent and this could be improved. Concerning gene symbols, there seems to be the habit that they are less and less frequently explained in recent publications; partly justified because the gene names are very long. Nevertheless, I think key gene symbols (such as CDH1, e-cadherin or cadherin-1) that are critical for a manuscript and its storyline should be explained. Many other terms are never properly explained throughout the manuscript, for example, also AJCC7, IGCA, and AJCC8. Its straightforward to pick up those names by going to the references given at these occurrences, but still... its just too many abbreviations. This may sound like little details but there are so many of them throughout the manuscript that some readers may get disturbed.
In the materials & methods section, there are also many little things missing, which make me believe that the manuscript was not prepared with the proper care and diligence; for example, the antibodies use for IHC are not fully disclosed (they are from SantaCruz; thats all we are told). These are just details but they indicate to me that the text needs to be more precise.
The expression levels of AhR and some of its target genes are compared in Table 2, providing some quantitative differences in mRNA expression. Does any of this correlate with differences observed in the immune staining of the corresponding tissues? This is particularly important as we all know that mRNA expression doesn't necessarily predict simultaneous changes in protein expression, and certainly nothing about subcellular localization and functionality.
Table 3 is very complex, and frankly speaking - quite boring to look at. Why dont the authors prepare some graphs highlighting just the most significant findings. This can also be presented in the form of tot plots, or ox plots; whatever, anything visual helps to apprehend the core findings more readily. Otherwise, table 3 is tedious to digest.
The authors have used only 1 cell line, KATO III, and that may not be sufficient as most findings in rigorous scientific papers are now presented in at least 2 or 3 cell lines, thus (ideally) showing some internal verification. Why was only 1 cell line used here? Is AhR not expressed in other GC cell lines? Differences in AhR expression and activity levels between cell lines may also add some very significant biological functionality correlations. (Worth trying, at least).
This would be particularly important as the KATO III cell line shows only some marginal changes for some of the genes, apart from CYP1A1 and IL1B. This may be confirmed, or even better, more substantiated if other cell lines were included.
Similar issues as pointed out with Table 2, also apply to Table 3. Its rather difficult to digest, why not extract the most differential findings and lot them as graphs. The table as such is just too big and too complex to get the message across. While plotting the data, it can also be attempted to group the candidates according to alleged functions, e.g., in EMT. Furthermore, genes that were investigated but didnt show any differences, could be eliminated or relegated to supplemental data.
Sometimes, data like those in Table 3 are also presented as heatmaps; that could also be tried.
Last not least, why not add any FUNCTIONAL tests to the purely gene-expression based data setss shown in this manuscript. Especially, issues like tumour cells motility (which may correlate with an EMT) are relatively simple to investigate, for example, in scratch wound assays. Maybe KAOT III cells display strong cell motility even in simple 2D monolayer cultures on plastic plates; then that would be easily accomplished.
Generally, I would strongly appreciate ANY functional validation that confirms ANY of the functions that are suspected to be involved; this would make the manuscript much stronger and would provide convincing data that justify acceptance and publication of this manuscript.
Comments on the Quality of English Language
the English language is not too bad; but there are some other issues which do not comply with the general practices used in scientific publications, these are outlined in the comments to the authors.
Author Response
Reviewer #3
Comments and Suggestions for Authors
The authors really like PCR. This manuscript is mainly driven by gene expression analyses, mostly by PCR, and to some degree, minor validation by immune histochemistry, but not in the cell line used for PCR, but in tissues. Thus, there is a lack of convincing functional data, although many of these are discussed broadly in the last section ("discussion")... but without much proof based on own experiments. Rather, based on publications by others.
Thus the manuscript lacks a bit of genuine novelty and originality, it starts addressing some issues like EMT and tumor cell invasion, but it doesn't even attempt to validate any of this in the only cell line that is used here. Using more than just one cell line, and adding at least SOME functional validation, would also help to bridge the conceptual gap between exploring tumor tissues (in the 1st half of the manuscript) and performing some experiments with a single cell line (Kato III).
This is in my opinion, insufficient to warrant acceptance and publication, and some validation on the cellular level, and NOT based on PCR, should be added.
Here are our reply to Reviewer 3’s general comments
Use of only one cell line: … this in the only cell line that is used here…. Using more than just one cell line….
The reviewer has missed that we actually used 2 different GC cell lines (Kato III and AGS) to validate the impact of POPs on AhR and AhR-related genes. We have now merged Figure 2 and Figure 3 into new Figure 2 to make it clearer. Further, in line, we normalized the Y axis in former Figure 3A (now new Figure 2b).
Functional validation proliferation, migration and invasion: …, it starts addressing some issues like EMT and tumor cell invasion, but it doesn't even attempt to validate any of this in the only cell line that is used here.
We would like to point out that such data are already reported in the literature. As examples, studies show that the AhR agonist TCDD (dioxin) promotes GC cell migration and invasion (Peng T-L, BMC Biology, 2009) and GC cell proliferation (Peng, T-L, 2009, World J of Gastroanterology). Thus, our data in humans are fully in line with these observations. We hope it will reassure the reviewer on the validity of our observations.
Use of primary tumor samples vs GC cell lines:… The authors really like PCR. This manuscript is mainly driven by gene expression analyses, mostly by PCR, and to some degree, minor validation by immune histochemistry, but not in the cell line used for PCR, but in tissues.
Yes, indeed, our study mainly focuses on primary GC tumors. We believe this is the strength of our study. We evaluate mRNA expression levels of various genes out of freshly-frozen samples of tumors compared to paired normal tissues.
Even though, we used Kato III and AGS cell lines as models to study how POPs impact on AhR and AhR-related gene mRNA expression, a 2D-study in such human cancer cell lines isolated decades ago and cultured in various conditions is far from reflecting what could have happened in GC patients after long-term exposure to POPs.
There are also many smaller issues listed below:
- …Concerning gene symbols, there seems to be the habit that they are less and less frequently explained in recent publications; partly justified because the gene names are very long.
The abstract could be somewhat streamlined, as it's complicated to read with all of the gene names or symbols; few of them are explained as it usually is in an abstract. Maybe the abstract should focus more on the main findings and therefore would also be less complex.
As stated by the reviewer, it is accepted to include gene symbols following their international code at the highest level of publication. Nonetheless, to please the reviewer, we have now indicated the full gene name before its international code for most, if not all cited genes, all along the manuscript.
Further, following the reviewer’s recommendations, we have now made numerous changes in the abstract (including expansion of the name of genes, pollutants, etc. We've also reworked the abstract to make it clearer. We hope that this new version will be fully appreciated by the reviewer.
- Also, the explanation of abbreviations is sometimes inconsistent in the manuscript, for example, POP is used before it is explained a few lines further down. In contrast, other abbreviations like TCCD and PCBs arent explained at all. Especially, some of the chemicals (like BaP), or chemical classes, are never properly introduced. Sometimes they are explained many pages further down (like TCCD). It would help many of the readers to disclose those names, especially in introduction, at first occurence, and this is also the habit in academic publications.
Generally, abbreviations and gene symbols are sometimes explained, sometimes not. Its just inconsistent and this could be improved.
Concerning gene symbols, there seems to be the habit that they are less and less frequently explained in recent publications; partly justified because the gene names are very long. Nevertheless, I think key gene symbols (such as CDH1, e-cadherin or cadherin-1) that are critical for a manuscript and its storyline should be explained. Many other terms are never properly explained throughout the manuscript, for example, also AJCC7, IGCA, and AJCC8. Its straightforward to pick up those names by going to the references given at these occurrences, but still... its just too many abbreviations. This may sound like little details but there are so many of them throughout the manuscript that some readers may get disturbed.
Following the reviewer’s recommendations, we went thoroughly and carefully all along the manuscript to explain all the abbreviations at first occurrence. Similarly, all chemicals are now properly introduced at the right place.
Further, in line with the reviewer’s comments, we have now indicated the full name before its international code for most, if not all, cited genes, throughout the manuscript.
We hope this great deal of work to follow the reviewer’s recommendations will be appreciated.
- In the materials & methods section, there are also many little things missing, which make me believe that the manuscript was not prepared with the proper care and diligence; for example, the antibodies use for IHC are not fully disclosed (they are from SantaCruz; thats all we are told). These are just details but they indicate to me that the text needs to be more precise.
We are really sorry to read that the reviewer thinks that our manuscript was not prepared with care. This is absolutely not the case. The material and methods section was written with care and in line with previously published studies from our laboratory, and we believe it provides all the required information to the readers, except indeed the reference for the two antibodies used for immunohistochemistry staining that indeed are missing. The precise references for those two antibodies have now been added.
- The authors have used only 1 cell line, KATO III, and that may not be sufficient as most findings in rigorous scientific papers are now presented in at least 2 or 3 cell lines, thus (ideally) showing some internal verification. Why was only 1 cell line used here? Is AhR not expressed in other GC cell lines? Differences in AhR expression and activity levels between cell lines may also add some very significant biological functionality correlations. (Worth trying, at least).
This would be particularly important as the KATO III cell line shows only some marginal changes for some of the genes, apart from CYP1A1 and IL1B. This may be confirmed, or even better, more substantiated if other cell lines were included.
We would like to point out that we actually used 2 different GC cell lines (Kato III and AGS) to validate the impact of POPs on AhR and AhR-related genes (see former Figure 3). We have now merged Figure 2 and Figure 3 into new Figure 2 to make it clearer.
- The expression levels of AhR and some of its target genes are compared in Table 2, providing some quantitative differences in mRNA expression. Does any of this correlate with differences observed in the immune staining of the corresponding tissues? This is particularly important as we all know that mRNA expression doesn't necessarily predict simultaneous changes in protein expression, and certainly nothing about subcellular localization and functionality.
This is a good point. Immunohistochemistry staining shows a strong increased expression of AhR and CYP1B1, just as what was seen by qRT-PCR. Thus, AhR and CYP1B1 mRNA expression levels are fully in line with protein expression.
- Table 3 is very complex, and frankly speaking - quite boring to look at. Why dont the authors prepare some graphs highlighting just the most significant findings. This can also be presented in the form of tot plots, or ox plots; whatever, anything visual helps to apprehend the core findings more readily. Otherwise, table 3 is tedious to digest.
Similar issues as pointed out with Table 2, also apply to Table 3. Its rather difficult to digest, why not extract the most differential findings and lot them as graphs. The table as such is just too big and too complex to get the message across. While plotting the data, it can also be attempted to group the candidates according to alleged functions, e.g., in EMT. Furthermore, genes that were investigated but didnt show any differences, could be eliminated or relegated to supplemental data. Sometimes, data like those in Table 3 are also presented as heatmaps; that could also be tried.
When a lot of data are generated, fully analyzing data takes time, what so ever it is presented in tables, boxplots, heatmaps, …
We may disagree with the reviewer, but tables allow to group a lot of information, and further, allow direct comparations between results for optimal interpretation of the data. We would like to kindly point out to the reviewer that ,for example, a boxplot representation of the data included in Table 3, would lead to the generation of 81 boxplots which I am sure he/she will recognize that it is not a way to simplify the interpretation of data. Altogether, this is why the presentation of all these date in Tables is a deliberate choice.
We also would like to emphasized that many other studies have selected the same optimal way to present similar large set of data, several of them being published in journals from MDPI. To cite some of our previous studies: Perrot-Applanat, M. et al., Biomedicines, 2022; Perrot-Applanat, M. et al., Cancers, 2022, Perrot-Applanat, M. et al., Oncol Lett, 2019; Vacher, S. et al., PLoS One, 2018,….
- Last not least, why not add any FUNCTIONAL tests to the purely gene-expression based data setss shown in this manuscript. Especially, issues like tumour cells motility (which may correlate with an EMT) are relatively simple to investigate, for example, in scratch wound assays. Maybe KAOT III cells display strong cell motility even in simple 2D monolayer cultures on plastic plates; then that would be easily accomplished.
Generally, I would strongly appreciate ANY functional validation that confirms ANY of the functions that are suspected to be involved; this would make the manuscript much stronger and would provide convincing data that justify acceptance and publication of this manuscript.
As indicated above in our general reply, we would like to point out that such functional data are already reported in the literature. As examples, studies show that the AhR agonist TCDD (dioxin) promotes GC cell migration and invasion (Peng T-L, BMC Biology, 2009) and GC cell proliferation (Peng, T-L, 2009, World J of Gastroanterology). Thus, our data in humans are fully in line with these observations. We hope it will reassure the reviewer on the validity of our observations.
We hope that you and the reviewers will now find this new version of our manuscript to be suitable for publication in Biomedicines
Sincerely,
Martine Perrot-Applanat and Véronique Baud
Round 2
Reviewer 1 Report
Comments and Suggestions for Authors
The authors revised their manuscript as previous comments. However, there are still a minor flaw that need to be revised. Reading reference 1 (CA Cancer J Clin 474 2024, 74(3):229-263.), it appears that gastric cancer is the fifth-most common malignancy and the fifth leading cause of cancer-related deaths, but the statement in the text has not been changed. They should revise it appropriately.
Author Response
The authors revised their manuscript as previous comments. However, there are still a minor flaw that need to be revised. Reading reference 1 (CA Cancer J Clin 474 2024, 74(3):229-263.), it appears that gastric cancer is the fifth-most common malignancy and the fifth leading cause of cancer-related deaths, but the statement in the text has not been changed. They should revise it appropriately.
We thank the reviewer for such a thorough reading of our manuscript. Following the reviewer’s comment, we have now modified the text indicating “fifth leading cause of cancer mortality, and not anymore “fourth”.
Reviewer 2 Report
Comments and Suggestions for Authors
Most issues have been stressed. However, the readability of Table 4 is still low. I still suggest visualizing Table 4 with plots. Though there will be plenty of boxplots as mentioned by the authors, only the ones with promising r values and significant p values can be kept in the main figures, leaving the others in the supplement. Also, multiple plots can be integrated into one big figure with various panels. It will make the results more vivid and attractive.
Author Response
Comments and Suggestions for Authors
Most issues have been stressed. However, the readability of Table 4 is still low. I still suggest visualizing Table 4 with plots. Though there will be plenty of boxplots as mentioned by the authors, only the ones with promising r values and significant p values can be kept in the main figures, leaving the others in the supplement. Also, multiple plots can be integrated into one big figure with various panels. It will make the results more vivid and attractive.
We are pleased that the reviewer has appreciated the great deal of work that we have made to improve the manuscript.
As stated in our previously reply to the reviewer, many previous studies, included ours, presented comparable data sets using tables. Presentation of data in a table in a proactive choice of ours.
Further, we would like to point out that the data presentation in table 4 was not at all a problem for the 2 other reviewers.
Therefore, especially in this summer break period during which a lot of personal is not at work, it will not be possible for us to change the way our data are presented in Table 4.
Reviewer 3 Report
Comments and Suggestions for Authors
The authors have addressed all the little issues like dozens of missing or unclear or disconnected abbreviations of genes that were pointed out in round 1, and also significantly improved the abstract which is now a lot more clear and easy to understand.
I am sorry that I have missed that the authors used more than one cell line in their validation studies (AGS in addition to KATO III). That was one of my points of criticism. Nevertheless, I still is not easy to spot especially in Figure 2, that 2 cell lines were used. Maybe the names of the lines could just be added to the figures themselves, not only the figure legends. This is a minor point.
The authors also have pointed out that tumor cell motility or invasiveness (as a sign for EMT?) is modulated by stimulating AhR, and that this has been previously published by other groups, as a reason not to perform matching validation studies themselves. I think this has now been better stressed in the new version and the references stating this are cited in the manuscript.
They focus instead on the correlations of AhR/AhRR/CYP1A1/CYP1B1 expression with "36 genes involved in EMT, cell proliferation, migration, immunity, and angiogenesis" in gastric cancers. That is of course more or less arbitrary cherry-picking based on gene signatures, or just a few components selected from larger signatures, and (in my opinion) not very informative. But if editors and the other reviewers find this sufficient and acceptable, its fine. It depends on how high the stakes for rigorous data sets shown in publications are put.
Author Response
The authors have addressed all the little issues like dozens of missing or unclear or disconnected abbreviations of genes that were pointed out in round 1, and also significantly improved the abstract which is now a lot more clear and easy to understand.
We are pleased that the reviewer has appreciated the great deal of work that was made to improve the manuscript, following at best the reviewer’s comments.
I am sorry that I have missed that the authors used more than one cell line in their validation studies (AGS in addition to KATO III). That was one of my points of criticism. Nevertheless, I still is not easy to spot especially in Figure 2, that 2 cell lines were used. Maybe the names of the lines could just be added to the figures themselves, not only the figure legends. This is a minor point.
Following the reviewer’s comment, we have now included the name of the two GC cell lines in Figure 2.
The authors also have pointed out that tumor cell motility or invasiveness (as a sign for EMT?) is modulated by stimulating AhR, and that this has been previously published by other groups, as a reason not to perform matching validation studies themselves. I think this has now been better stressed in the new version and the references stating this are cited in the manuscript.
We thank the reviewer for such a positive appreciation.
They focus instead on the correlations of AhR/AhRR/CYP1A1/CYP1B1 expression with "36 genes involved in EMT, cell proliferation, migration, immunity, and angiogenesis" in gastric cancers. That is of course more or less arbitrary cherry-picking based on gene signatures, or just a few components selected from larger signatures, and (in my opinion) not very informative. But if editors and the other reviewers find this sufficient and acceptable, its fine. It depends on how high the stakes for rigorous data sets shown in publications are put.
We believe our study brings novel valuable data on the role of AhR and AhR-related genes in gastric cancer. Such opinion appears to be shared by the two other reviewers.
Further, we would like to point out that all data were generated in a sound and rigorous manner.